

# Detection and estimating the blood accumulation volume of brain hemorrhage in a human anatomical skull using a RF single coil

Moshe Oziel[1], Boris Rubinsky[2] and Rafi Korenstein[1]

[1] Department of Physiology and Pharmacology, Tel-Aviv University, Tel-Aviv, Israel
[2] Department of Mechanical Engineering, University of California, Berkeley, Berkeley, CA, USA

## ABSTRACT

**Objective:** An experimental study for testing a simple robust algorithm on data derived from an electromagnetic radiation device that can detect small changes in the tissue/fluid ratio in a realistic head configuration.

**Methods:** Changes in the scattering parameters ($S_{21}$) of an inductive coil resulting from injections of chicken blood in the 0–18 ml range into calf brain tissue in a human anatomical skull were measured over a 100–1,000 MHz frequency range.

**Results:** An algorithm that combines amplitude and phase results was found to detect changes in the tissue/fluid ratio with 90% accuracy. An algorithm that estimated the injected blood volume was found to have a 1–4 ml average error. This demonstrates the possibility of the inductive coil-based device to possess a practical ability to detect a change in the tissue/fluid ratio in the head.

**Significance:** This study is an important step towards the goal of building an inexpensive and safe device that can detect an early brain hemorrhagic stroke.

# INTRODUCTION

The chance of having a hemorrhagic stroke is between 0.01% and 0.03% of the worldwide population (*Qureshi, Mendelow & Hanley, 2009*). The risk of dying a year after a stroke is about 50% (*An, Kim & Yoon, 2017*). Studies have shown that the volume of bleeding in the brain increases within the first 3 h of a stroke for 73% of patients (*Davis et al., 2006; Brott et al., 1997*) where an increase in bleeding volume is closely linked to lower chances of recovery from the stroke (*Davis et al., 2006; Dowlatshahi, 2011*). Thus, precise diagnosis and treatment swiftness seem to be important factors in patient survival (*Brain Aneurysm Foundation, 2019*).

When a symptomatic patient is brought to a hospital, the hemorrhage can be detected by Magnetic Resonance Imaging (MRI), Computed Tomography (CT), or Doppler Ultrasound (*Huisman, 2005*). These imaging technologies are accurate and supply the relevant information needed for diagnostics. However, the price of these devices is high and require skilled operators. In the developing world, most of the population has no

Corresponding author
Moshe Oziel,
mosheoziel@mail.tau.ac.il

access to these technologies (*González, Blumrosen & Rubinsky, 2010*). While CT is commonly used for detecting a hemorrhagic stroke, the possibility to detect temporal hemorrhage is limited to the number of repeat CT scans allowed, in particular for children who are most susceptible to the X-ray radiation employed by the CT (*Little, 2008*).

A measure for the occurrence of progressive internal bleeding is to detect if the volume ratio of tissue/fluid in an organ changes over time. We have developed several simple, non-contact, and inexpensive radiofrequency (RF) and microwave (MW) frequency based diagnostic technologies for detecting internal bleeding in the body, with a goal towards their use for the economically disadvantaged populations. One of these technologies, referred to as volumetric integral phase-shift spectroscopy (VIPS), monitors the change in the amplitude and phase between two coils across the head in the frequency range of RF (*González & Rubinsky, 2006*). In a clinical study with 248 subjects, VIPS has shown that it can detect a severe stroke in seconds with a 92% accuracy (*Kellner et al., 2018*) and has been chosen as one of the top ten medical innovations for 2019 (*Cleveland Clinic, 2019*). VIPS has received FDA and CE approval and is used clinically to detect brain hemorrhages (*Gonzalez et al., 2014*), cerebrovascular autoregulation monitoring (*Oziel et al., 2016*), or to detect fluid shifts in the brain during dialysis (*Rao, 2018*). There are many other uses of RF/MW measurements for diagnostics in medicine. For example, for monitoring breathing (*Richer & Adler, 2005*; *Teichmann et al., 2013*, *2014*), magnetic imaging tomography (MIT) (*Mobashsher & Abbosh, 2014*) or thermal therapy monitoring (*Haynes, Stan & Moghaddam, 2014*). The technologies that we have developed emerge from the fact that the intracranial volume is constant, and changes in the ratio of blood/brain tissue structure alter the bulk electromagnetic properties of the entire head (*Griffths, Stewart & Gough, 1999*; *Munawar Qureshi, Mustansar & Mustafa, 2018*).

The VIPS technology employs a transmitter coil and one or more receiver coils. While precise, this makes the technology sensitive to motion artifacts, given the major contribution to the signal strength is usually the direct path between the transmitter and receiver. One possible solution is using a single coil, thereby removing the location dependance between two coils. Due to the spatial symmetry of the induction coil (surrounds the head), there is no single path with an extraordinary contribution to the received signal. Therefore, the response to the relative change in the position of the single coil will be less significant than the response to the change in the relative position in a two-coil system. Our previous studies of monitoring the brain by a single coil began with mathematical modeling. We used numerical simulations to analyze the performances of a spiral antenna (*Oziel, Korenstein & Rubinsky, 2017*) and a single inductive coil surrounding the head (*Oziel, Korenstein & Rubinsky, 2018*). These feasibility studies evaluated the sensitivity of the radar-based measurements to small changes in the blood/tissue volume ratio in the case of a brain hemorrhage. The simulations predicted that these devices can detect changes as small as two ml in the blood volume. We also found that a device based on a single-coil is superior to the spiral antenna when the bleeding location is unknown. In a further evaluation of the technology, we concluded that the single inductive coil is easier and less expensive to manufacture and use than the spiral antenna configuration (*Oziel, Korenstein & Rubinsky, 2018*). Next, we performed an

experiment on a cylindrically shaped alcohol-based phantom gel that simulates brain tissue, into which we injected a physiological fluid that simulates blood (*Oziel et al., 2020*). We also developed a new statistical method to analyze the experiments with this simple configuration. The analysis of these experiments shows that a single coil operating at 100–1,000 MHz can detect changes in the tissue/fluid ratio as small as two ml. We learned that there are only a small number of frequencies that are strongly affected by changes in the tissue/fluid ratio. These frequencies are affected mainly by boundary conditions as well as by the location of the accumulated liquid. We have also shown that for those frequencies, there is a correlation between the changes in amplitude and phase in response to the injection of physiological saline (*Oziel et al., 2020*).

Recently, we developed a very simple and robust algorithm for the single inductive coil configuration (*Oziel, Korenstein & Rubinsky, 2020*). The algorithm can detect changes in the liquid/gel ratio in the cylindrical configuration of the head (*Oziel et al., 2020*) with very high accuracy. This algorithm was also able to roughly estimate the injection volume.

In the present study, we validate the single-coil based technology of detection internal head bleeding conceived first in previous study (*Oziel, Korenstein & Rubinsky, 2017*), on a phantom as close as possible to a real human head. This is an experimental study on a human anatomical shape skull with real blood and brain tissue. We examined the performance of the detection algorithm (*Oziel, Korenstein & Rubinsky, 2020*) in this geometric and tissue configuration, which resembles more closely the clinical situation with respect to: (a) the sensitivity to small changes in the tissue/fluid volume ratio; (b) the effect of accumulated location of bleeding; (c) the difference between the response to blood injection into a balloon, which simulates the accumulation of blood in one place in the brain (such as in an extradural hematoma), in comparison to direct injection of blood into the brain tissue, such as occurs in an intraventricular hemorrhage.

## MATERIALS AND METHODS

### The experimental system

As mentioned before, the system design was justified using 3D simulations (*Oziel, Korenstein & Rubinsky, 2018*) and then by a simple model of gel and saline solutions representing brain and blood tissues, respectively (*Oziel et al., 2020*). The experimental system constructed around an Foxfield N9923A Network Analyzer (Keysight, Santa Rosa, CA, USA). The network analyzer was connected to the induction coil sensor by a single co-axial cable. The electrical circuit, photographs of the devices and $S_{11}$ diagram are shown in Fig. 1. A total of 1 mm diameter copper electric wire, used for transformer applications, was wrapped (72 loops) in a single layer around a plastic cylinder (width of 30 mm). The circular cross-section of the plastic cylinder possessed a 240 mm diameter, 60 mm height and a 1 mm thickness. The network analyzer was connected to a laptop, Lenovo W541 (Lenovo, Morrisville, NC, USA), through a LAN connection. The blood injection was performed by a KDS-210 Syringe pump (KD Science, Holliston, MA, USA) that can move in 0.165 micrometer step increments, equivalent to an injected volume of 0.092 μl per single step. The pump was connected to the laptop via an RS232 to

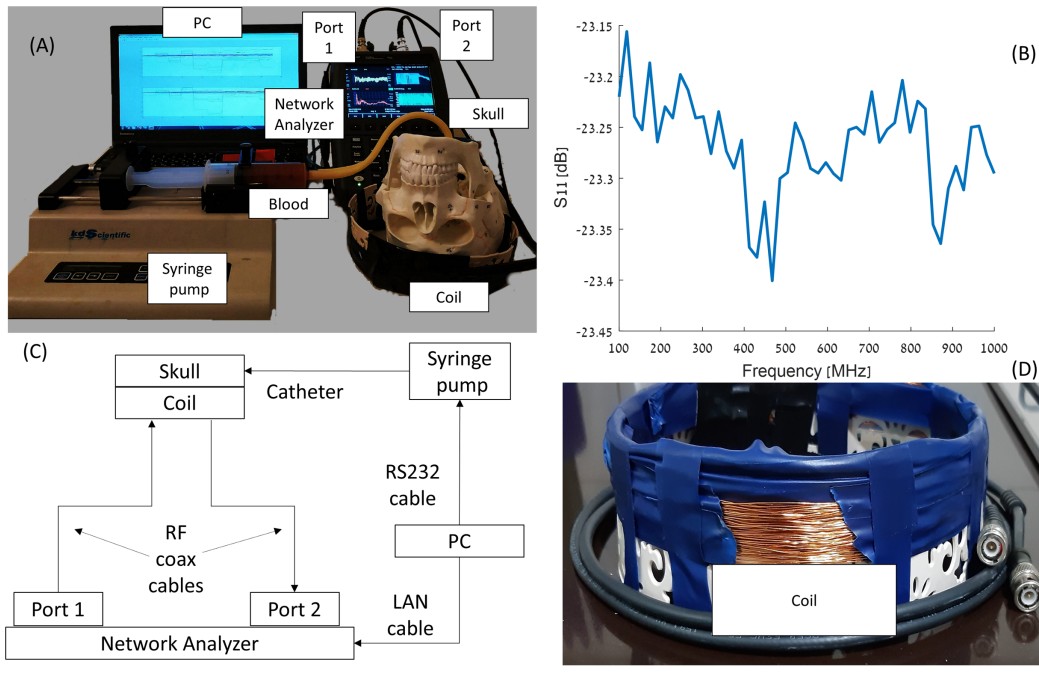

**Figure 1** Visualization of the elements in the experiment (A and D), $S_{11}$ diagram of the coil (B), and scheme of the experimental network (C).

RJ11 cable. The laptop controls the syringe pump and collects and stores the data from the network analyzer.

## The experimental model for simulating internal bleeding

Intracranial hemorrhages are divided based on their anatomic location and can be classified into two groups: intra-axial and extra-axial ones (*Caceres & Goldstein, 2012*). The intra-axial group possesses two types of hemorrhages. The first type is an intraparenchymal hemorrhage, which reflects bleeding within the brain parenchyma (*Almandoz et al., 2009*; *Gebel et al., 1998*). This type of hemorrhage has no distinctive shape and the range of blood volume is between 0 and 200 ml, with an average volume of 68.7 ml. The second type is intraventricular hemorrhage. An intraventricular hemorrhage displays a bleeding into the brain's ventricular system (*Gates et al., 1986*). The range of blood volumes is in the range of 0–20 ml (*Hallevi et al., 2009*; *Kramer et al., 2010*). The extra-axial group consists of three types: (i) An extradural hematoma (epidural) which is a lens-shaped collection of blood between the dura and the inner surface of the skull (*Pryse-Phillips, 2009*), within a 0–150 ml volume range (*Rivas et al., 1998*); (ii) A subdural hematoma is a crescent-shaped collection of blood between the dura and the arachnoid (*Pryse-Phillips, 2009*). The average volume of the subdural hematomas is 91 ml in the range of 5–300 ml (*Gebel et al., 1998*) (iii) Subarachnoid hemorrhage is bleeding between the arachnoid membrane and the pia mater (*Pryse-Phillips, 2009*). The typical shape for a subarachnoid hemorrhage on CT images is a so-called "star sign".

In the literature, one can find different types of experimental models that simulate the brain tissue, depending on the application and the measuring instrument. For RF

applications, saline solution was used at different concentrations to detect small changes in conductivity at 3.5 GHz (*Karanasiou & Uzunoglu, 2004*). A more recent study used a gel with the dielectric properties of the brain's white matter to detect small changes in conductivity through radiometry (*Groumpas et al., 2017*). Nearly two decades ago a more advanced model consisting of a uniform phantom based on COST244 to create tissue-equivalent phantoms using mixtures of agar, polyethylene powder (PEP), TX-151, and sodium chloride was used to perform SAR estimates at 200–3,000 MHz frequencies (*Okano et al., 2000*). Using identical materials (*Velander et al., 2018*) constructed a phantom to assess intraocular pressure. A good match between the dielectric parameters of the brain and the tissue of water-based agar and sugar at different doses was shown (*Chew et al., 2012*). A path-loss index, using a square phantom based on the dielectric parameters at a single 2.4 GHz frequency was measured (*Roelens et al., 2006*).

Using a whole animal or animal tissue for human modeling is also common in research. *Lind et al. (2007)* presents a summary of dozens of uses of pig tissue in neurological research. *Shaver et al. (1996)* used three piglets to model brain injury after a subdural hematoma. *Fantini et al. (1999)* used pigs and light spectroscopy to detect disturbances in brain hemodynamics, and *Zhou et al. (2009)* performed similar research. *Sanchez (2013)* examined in his thesis how the injection of blood into sheep brain tissue affects the signal return of an antenna in a broadband pulse. *Ayati, Bouazza-Marouf & Kerr (2014)* researched a way to find the location of a hematoma in a sheep brain via antenna array and impedance tomography.

Since the head size and anatomy of the various in vivo animal models differs significantly from that of the human head, it will affect the estimation of volume of blood accumulation. Therefore, we have chosen to employ in our experiment a model of human skull (Wellden, Toronto, ON, Canada), prior to performance of clinical trials. The skull we used in the experiment is based on Medical Anatomical Human Skull model, made of PVC in a realistic adult size. The length of the skull was 17 cm and the width were 13.5 cm (on the semi minor/major axes). The volume of the brain cavity was approximately 1,200 cm$^3$ (see Fig. 2). We inserted into the skull cavity a thin plastic bag, which contained parts of defrosted calf brain (originally prepared for food). For each experiment, we verified that the volume of the brain filled the skull cavity almost completely. After inserting the brain into the skull, we filled all the remaining space with physiological saline solution. Hemorrhage was simulated by injecting defined volumes of blood into the brain tissue using a standard 2-way Foley catheter with a rubber valve (Bard, Covington, GA, USA). The positions of catheter tips are marked in Fig. 2. We roughly divided the injection locations into two groups, relative to the center point. One injection site was at the skull center ±1 cm (blue circles, Fig. 2) and second injection sites were at the margin of the skull (the orange circles—about 1 cm from the outer surface ± 1 cm). The height of the catheter's tip was ~2 cm from the top of the skull ±1 cm. In the experiment, we used fresh chicken blood from a chicken slaughtered for food 2 h before the experiment. After an hour of experimentation, we discarded the blood (we did not use anticoagulants). The blood was injected through two routes. In the first one, the blood was injected into a rubber balloon capable of reaching a volume of 70 ml.

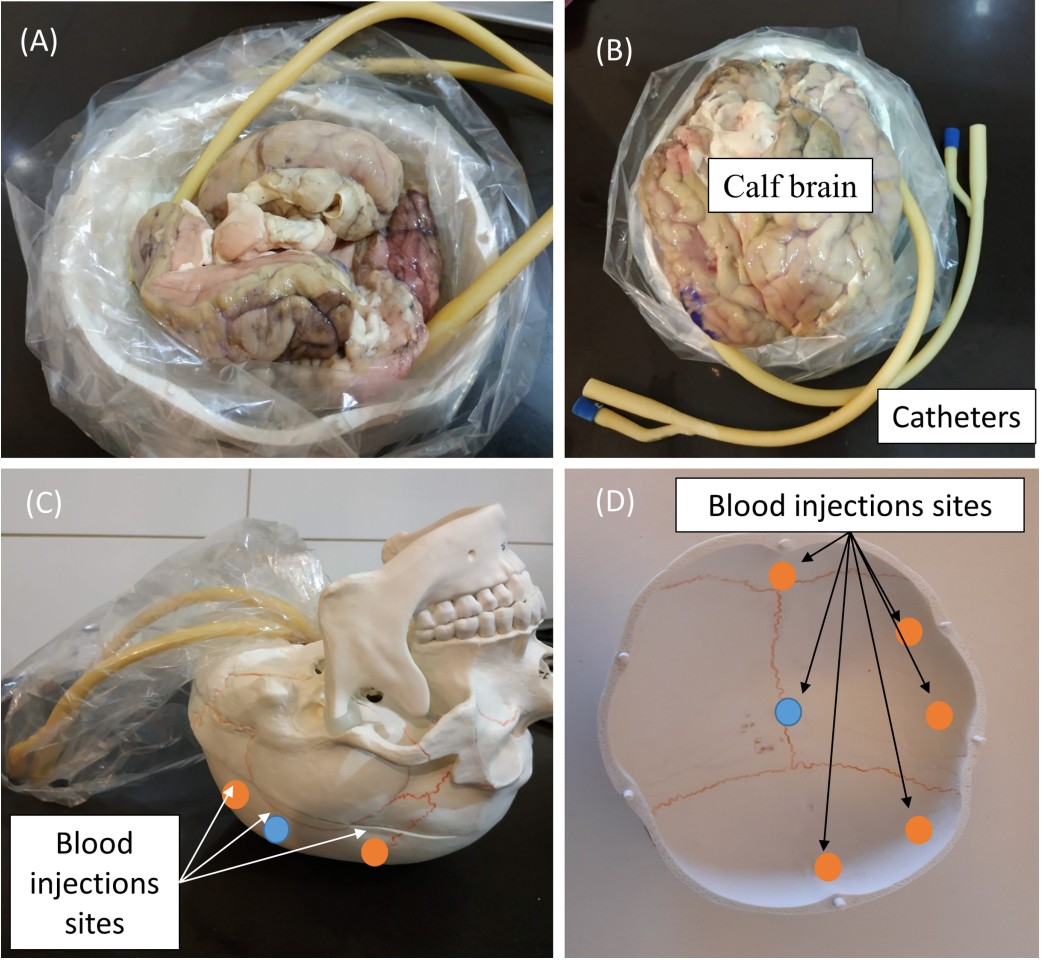

**Figure 2 Steps in preparing the skull for an experiment (A and B), and the blood injections sites (C and D).**

The balloon was connected with epoxy glue to the tip of the catheter (standard catheter balloons have a volume of only 10 ml). The second route was a direct blood injection into the brain tissue on the same injection sites as the first route. In both cases, we used a 50 ml syringe (MedicPro, Kuala Lumpur, Malaysia), which was controlled by a syringe pump and was connected directly to the catheter's tube.

## Experimental method

We used a network analyzer to measure $S_{21}$—the power received in the network analyzer output port (port 2) relative to the power input port (port 1). We measured 50 equally spaced $S_{21}$ ranged from 100 MHz to 1 GHz per 600 ms with no averaging. The bandwidth was 300 Hz and maximum power level was 5 dBm, continuously throughout the experiment. Hemorrhage was simulated by injecting precise volumes of chicken blood into the brain tissue (with/without balloon) in two ml increments. For every injected volume increment, we took 70 $S_{21}$ measurements during a 1 min period. The injected volumes ranged from 0 to 18 ml (9 injections).

In this experiment, we examined several parameters. As we previously described, we roughly divided the blood injection locations into those made in the center of the skull and those made at its margins. Another parameter is injection into a balloon, which intends to simulate blood accumulation in a defined site in the brain, or blood injection directly into the brain tissue, which is supposed to simulate a more delocalized brain hemorrhage. Finally, to test the effect of blood pulsatility, we performed measurements in which we cyclically pushed and pulled one ml of blood approximately 10 times per minute. These measurements were designed to test whether our algorithm was able to deal with periodic changes that generate noise in a different way than noise originating from an electronic source. Obviously, the heartbeat rate is much higher than the pump rate, but in our experiment the Nyquist frequency is about 30 Hz. So high frequencies of this frequency will not be properly sampled.

To reduce RF reflection from the environment, a cylindrical Faraday cage possessing with diameter of 250 mm and height of 120 mm, was placed around the inductive coil sensor. We have not maintained a fixed position of the coil and the phantom relative to the cage. So, between each measurement, the relative positions of the phantom, the cage, and the coil have slightly changed.

We estimated the internal errors, such as thermal noise and internal drift in the same method as in the previous experiments (*Oziel et al., 2020*): 20 min of the $S_{21}$ measurements under 3 different scenarios: (a) The device was turned on for half an hour; (b) when the device was turned on for 2 h (hot device); and (c) in the presence and absence of the skull with the brain. Three repeats were done for each scenario. After verifying that the noise distribution was normal, we calculated the average over the different measurements and the standard deviation for both the amplitude and phase. We could estimate the relative error due to noise was $2.1 \times 10^{-4}$ dB in the amplitude measurements and $1 \times 10^{-3}$ degree in the phase measurements.

Changes in the relative position of the faraday cage during measurement have a strong effect on the electrical signal, which is fundamentally different from changes resulting from the injection of blood into the skull, which is expressed in the form of high and narrow peaks over the entire frequency range. Measurements in which a relative change in the location of the cage was detected were rejected.

## Algorithm

The description of the algorithm used for the analysis of the results is described as follows:

1. Let us define $t_{start}$ and $t_{end}$ as the start and the end time points at which the phase and amplitude are measured across the RF coil. In our experiment there is actually a correlation between time and the injected volume, since we injected in increments of two ml of blood volume, every single minute. However, the algorithm described here is general.

2. Let us define $F_0$ as the analyzed frequency range, where n is the number of frequencies sampled in the $F_0$ frequency range. In our experiment, $F_0$ is in range of 100–1,000 MHz and $n = 50$. In general, we require that $n \geq 30$.

3. The Network analyzer was used to record the amplitude, A, and the phase, Ø, at each frequency $f_0 \in F_0$, during the period of time from $t_{start}$ to $t_{end}$ (to avoid repetition, we will use the notation E, as a place holder for amplitude and phase). For each frequency, $f_0$, and time, $t$, we measure multiple values of E (70 samples in our experiments) in a predetermined narrow time window, $\Delta t$ to increase the signal to noise ratio (SNR) and calculate the average, $avg(E(f_0, t))$, and standard deviation, $std(E(f_0, t))$ for that time, $t$. Again, in this experiment $\Delta t$ corresponds to the 1-min time duration between each injected volume of blood. As before, the algorithm developed here is general and not related to the measurement time window.

4. The algorithm is based on measuring the changes in measurements of $E(f_0, t)$ between those made at $t_{start}$ and $t_{end}$.

5. As a first step, we filter out from all the measurement frequencies, all frequencies at which $|avg(E(f_0, t_{end})) - avg(E(f_0, t_{start}))| <$ Measurements error ($2.1 \times 10^{-4}$ dB for the amplitude and $1 \times 10^{-3}$ degree for the phase) and define $F_1$ as the set of all frequencies in the measured range in which the change is higher than the measurement error.

6. Let us define the ratio between the change in the average values at the two time points and the noise (Represented by the standard deviation of E) at each frequency, $f_1 \in F_1$ defined in step 5., as follows:

$$R(f_1) = \frac{|avg(E(f_1, t_{end})) - avg(E(f_1, t_{start}))|}{\max(std(E(f_1, t_{end})), std(E(f_1, t_{start})))} \tag{1}$$

where E (as defined in paragraph 3) can be the amplitude or the phase.

7. The next step is to filter out all frequencies in which $R(f_1) < 2$ and let's define $F_2$ as the set of all frequencies in the measured range at which $R(f_1) \geq 2$ (A dimensionless number). This should remove most of the noise-induced artifacts.

8. Then, for $f_2 \in F_2$ we examine whether the difference between $E(f_2, t_{end})$ and $E(f_2, t_{start})$ is statistically significant for each of the frequencies in the set $F_2$. To this end, we use the two-sample Kolmogorov–Smirnov test (*Lilliefors, 1967*), which examines whether the two groups, $E(f_2, t_{end})$ and $E(f_2, t_{start})$, belong to the same distribution. We then define the set of frequencies, $F$, in which the measurements satisfy the following properties: For each set of the measurements E in the set of frequencies, $F$, $R(f_2) \geq 2$ and the statistical test shows that there is a statistically significant difference between $E(f, t_{end})$ and $E(f, t_{start})$ where $f \in F$.

9. Finally, let us define $\overline{R(F)}$ as:

$$\overline{R(F)} = \frac{\sum R(F)}{n} \tag{2}$$

Here also, $\overline{R(F)}$ is a dimensionless number and is the average change in the measured amplitude or phase between two measurements across a time interval relative, to the standard deviation of the experimental measurements for all the statistically meaningful frequencies. Importantly, we divide the contribution of R(F) from all frequencies by the number of frequencies of $f_0$. Doing this, allows us to deal with edge cases where the

difference between $E(f, t_{\text{end}})$ and $E(f, t_{\text{start}})$ can be a consequence of noise from a small number of frequencies.

In the next paragraph, we will show that the value of $\overline{R(F)}$ is a very robust and accurate parameter that can serve as a marker for changes in blood volume. This can serve as a very simple indication of changes in the medical condition of the patient. It must be emphasized that this study was done employing a brain tissue model in a skull, and the algorithm must be clinically verified in the future.

## RESULTS

We have obtained a complete record of the changes in amplitude and phase measured across the RF coil in response to injection of blood in our experimental model of intracranial hemorrhages, throughout the frequency range of 100 MHz to 1 GHz. The results are qualitatively similar to those presented in our previous theoretical studies (*Oziel, Korenstein & Rubinsky, 2017*; *Oziel, Korenstein & Rubinsky, 2018*). We find that the change in amplitude and phase across the RF coil upon an injection of blood, is a complex function of frequency, of the injected blood volume and of the location. The frequency response to the injected blood is neither uniform across all frequencies nor monotonous with the increase in the injected blood volume. In fact, at some frequencies there are no changes with the injected blood volume, while at others the changes are substantial. Figure 3 illustrates the complexity of the frequency response to blood injection into the skull's center. The frequencies in this figure were chosen at random, just for demonstration purposes. Figure 3 shows the average changes in amplitude and phase due to nine injections of consecutive two ml volumes of blood. The left panel shows the change in the absolute value of the amplitude and the right panel shows the change in the absolute value of the phase. The abscissa is the volume of the injected blood, the blue squares markers are the average measurement values of $E$ for 559.18 MHz, the red circles markers are the measurement values of $E$ for 540.82 MHz, and the magenta asterisk markers are the measurement values of $E$ for 173.47 MHz.

It is evident from Fig. 3 that the magnitudes and nature of the change in amplitude and phase between consecutive injections of two ml volumes, are dependent on the frequency at which the measurements were made. For example, at 559.18 MHz and 540.82 MHz, the difference between the values of amplitudes and phase for consecutive injections of two ml volumes can be substantial. In contrast, at 173.47 MHz, the amplitude and phase do not vary significantly between consecutive injections. The frequencies of 559.18 MHz and 540.82 MHz are close to each other relative to the entire range of frequencies in this experiment. However, the response to the injection of two ml of blood is substantially different. The right panel shows that up to a total injected volume of 12 ml, the curve representing change of phase increases monotonically with the injection of consecutive volumes of two ml blood, for both frequencies, of 559.18 MHz and of 540.82 MHz. However, the magnitude of the change of phase remains constant for further injections of blood, for the frequency of 559.18 MHz, but not for the frequency of 540.82 MHz. In contrast, the change in amplitude due to injection of two ml of blood exhibits a

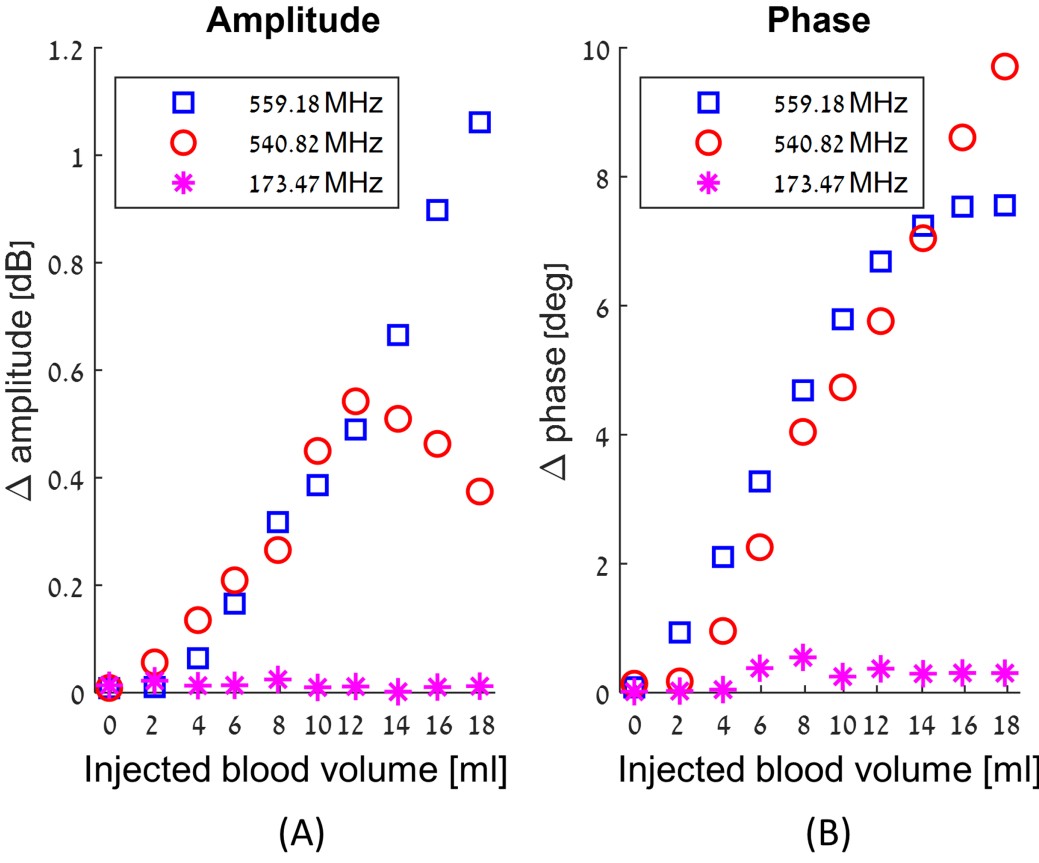

**Figure 3 Examples of the changes in amplitude (A) and phase (B) in response to incremental increase in the volume of injected blood at the center of the skull, as a function of three different arbitrary chosen frequencies.**

completely different pattern for these frequencies. At the frequency of 559.18, the value of the amplitude across the RF coil increases monotonically with the injected volume of blood. However, at a frequency of 540.82 MHz, the amplitude across the RF coil increases with an increase in the injected volume of blood, until an injected volume of 10 ml. At this frequency, the amplitude begins to dramatically decrease with any further injection of blood after a total volume of 10 ml blood was injected.

In summary, this example demonstrates that there is no trivial correlation between changes in the volume of blood injected into the brain and the phase and amplitude measurements across the RF coil. Obviously, changes in blood volume cause changes in the amplitude and frequency across the single coil RF. However, these changes are not trivial. For this technology to have practical value, there is a need for an algorithm that can analyze the entire frequency range and extract the appropriate information. We have introduced the algorithm in the Material and Methods section and will subject the experimental results in this study to this algorithm.

Histograms of all R(F) values in this experiment, for amplitude and phase, are shown on the left-hand side of Fig. 4. This kind of distribution can be transformed into a normal distribution by applying a $\text{Log}_{10}$ transformation (*Howell, 2007*). The $\text{Log}_{10}\left(\overline{R(F)}\right)$ values

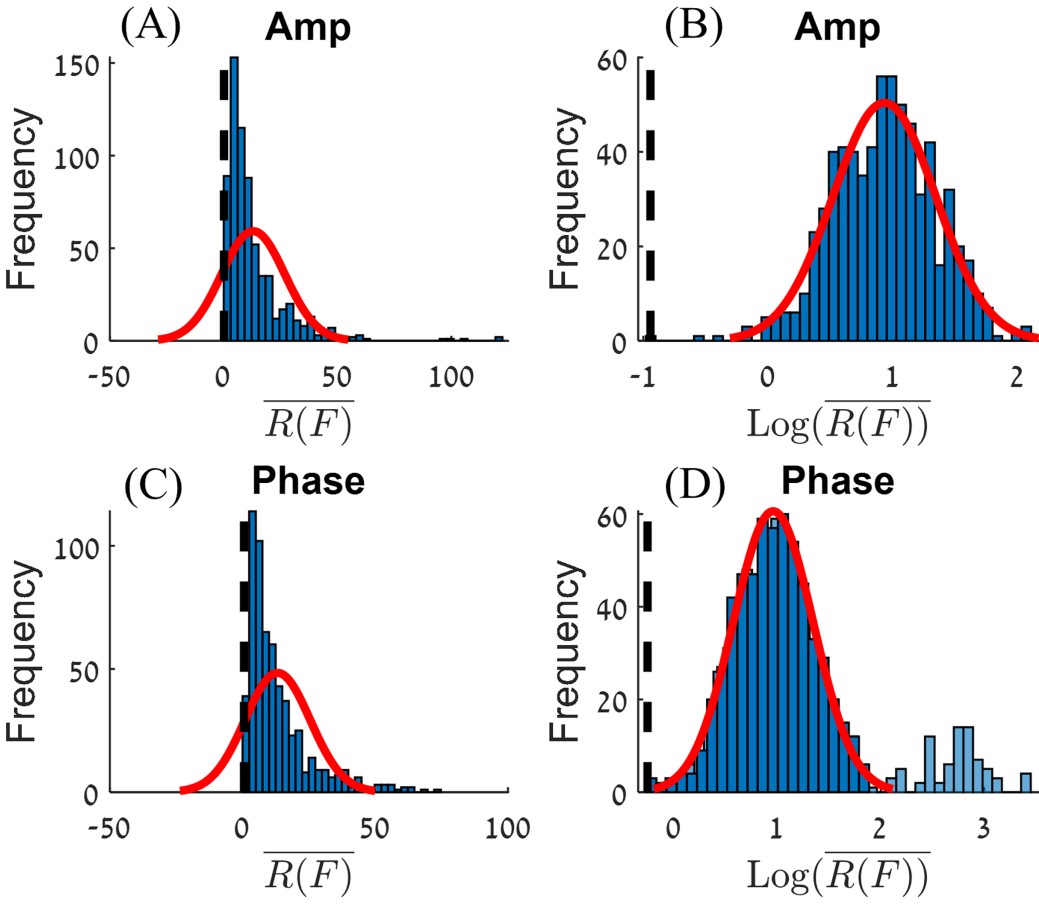

**Figure 4** **Histograms of R(F) (A and C) and Log$_{10}$(R(F)) (B and D) for the amplitude (A and B) and phase (C and D).** A normal distribution plot is shown for comparison (red line). The normal distribution of (D) had fitted on the data without the extreme values (the light blue bars).

for amplitude and phase are shown on the right-hand side panels of Fig 4. The center of the normal distribution for the amplitude is around $\overline{R(F)} = 15$. For the phase, there are extreme values for $\overline{R(F)}$, $(\overline{R(F)} > 100)$. After filtering out these extreme values (the criteria used to filter out the extreme values will be presented in the discussion section), we obtain a normal distribution where the center of the distribution for the phase is around $\overline{R(F)} = 11$.

The processed raw data from the experiments is given on Fig. 5. The figure shows the $Log_{10}\left(\overline{R(F)}\right)$ values for amplitude and phase, for each experimental condition, that includes variation in volume of injected blood and location of the injected volume.

Figure 5, show the $Log_{10}\left(\overline{R(F)}\right)$ values for amplitude and phase at a boxplot format, where the abscissa is the volume of injected blood. For example, the values at four ml represent the algorithm values for the difference between 0 and 4 ml, as well as the algorithm values for the difference between 2 and 6 ml, 4 and 8 ml, and so on. Each data point represents an experiment. The red line inside the blue box represent the median of each set of measurements and the blue box represent the interquartile range (25th to the

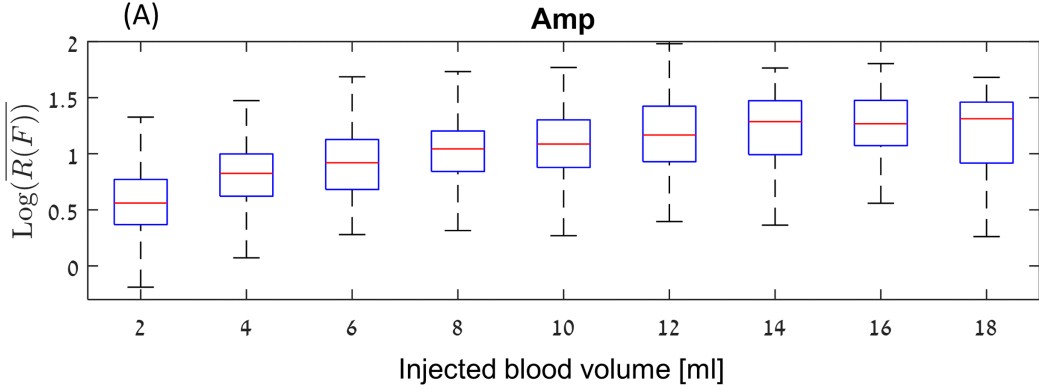

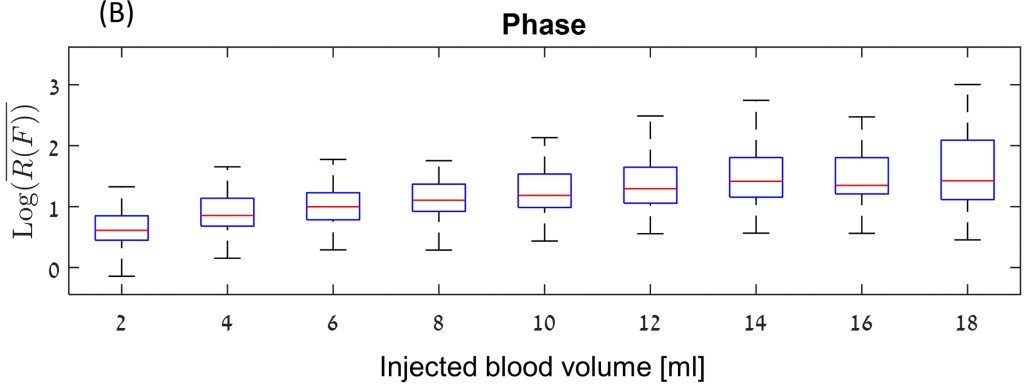

**Figure 5 Log$_{10}$ (R(F)) values for amplitude (A) and phase (B) at a boxplot format (without the outliers).**

75th percentile). This graph shows the following phenomena. In general, the change for the amplitude is between the signal-to-noise ratio of 3 ($\mathrm{Log_{10}}\left(\overline{R(F)}\right) \sim 0.5$) for low differences in the accumulated blood volume, to 15–50 for high differences between the accumulated blood volume, ($\mathrm{Log_{10}}\left(\overline{R(F)}\right) \sim 1.7$). The median values of the amplitude are lower than the phase values. However, the phase values include far more extreme values (a signal-to-noise ratio of 500–1,000) As expected, the median value increases depending on the injection volume up to around 14–18 ml. It appears that the effect of amplitude or phase for a 14 ml change in blood volume is not significantly different from the change in amplitude or phase for 16 or 18 ml.

Figure 6 shows the results of all the various experiments on a consolidated graph. The abscissa is $\mathrm{Log_{10}}\left(\overline{R(F)}\right)$ for the amplitude, the ordinate is $\mathrm{Log_{10}}\left(\overline{R(F)}\right)$ for the phase. The different markers represent blood injection volumes in the 2–18 ml range, and the colors represent the different experimental groups.

The green markers are for an experiment in which one ml blood was pushed into and out of the balloon in the brain tissue at a frequency of 0.17 Hz at the center of the skull. The group of experiments in which the blood was injected into a balloon (both center and margin) in increments of 2ml, is marked by blue. It is interesting to notice, that in most of the experiments there is a linear relationship for $\mathrm{Log_{10}}\left(\overline{R(F)}\right)$ between amplitude

values and phase values. However, in a small fraction of the results (5.2%) the change in phase is 10 times as large as the change in the amplitude. This phenomenon can be explained by the convergence of different boundary conditions, so that local enhancement is created for a specific injection (In the next paragraph, we will discuss the algorithm for estimating the volume of blood flow, where we will see a model that ignores these values. It is important to understand that these high values are strong evidence of an increase in blood volume in the brain, and from a clinical point of view, they are a trigger to alert of a significant change in blood volume in the brain). The correlation between $\text{Log}_{10}\left(\overline{R(F)}\right)$ is the same as that for blood injected in a balloon at both center and the margin sites. Again, in a small fraction of the measurements the change in phase values are larger than the change in amplitude values.

## DISCUSSION

Our study demonstrates that the monotonic increase in blood volume in the skull does not lead to a monotonic increase in the measured amplitude and phase change across the coil. We also observe that there are a small number of frequencies that are strongly affected by the change in the tissue/fluid volume ratio in the brain. However, the intensity of the change is different from frequency to frequency and from experiment to experiment. This uncertainty about which of the frequencies will produce a stronger response to changes in blood volume in the head and the lack of monotonicity in the response at many frequencies have suggested to us that the measurements should be made across a large range of frequencies, in the hope that this will capture the frequencies with a strong response. In this study we made the measurements across a range of frequencies from (100 to 1,000 MHz), although there is no technological impediment to making measurements across a larger range.

In the paragraph describing the algorithm, we define $\overline{R(F)}$ as the sum of the algorithm values, divided by the number of $f_0$ values—the number of frequencies scanned. This change allows us to give less weight to cases where, at a single frequency, there is a significant power change as a result of random noise (e.g., spike noise).

One can see in Fig. 6 that all the algorithm values (green markers) which arrived from the one ml group, have values less than 0.5, namely $\overline{R(F)}_{\text{amp/phase}} < 3$. This is in agreement with our previous article (*Oziel, Korenstein & Rubinsky, 2020*), in which we showed that this threshold value corresponds to a value that distinguishes small changes of noise and volume of blood from real significant changes of the tissue/fluid volume ratio in the head. A change in the blood/tissue ratio of $P\left(\text{Log}_{10}\left(\overline{R(F)}\right)\right) = 0.5$, which equals to $\overline{R(F)} = 3$ can be used as a measure for the occurrence (a diagnostic index) of a clinical change in the blood/tissue volume ratio.

Correctly estimating the volume of blood that accumulates in the brain is of clinical importance. We would like to estimate the volume of the injection depending on the values of R(F), amplitude and phase. For this purpose, we selected the set of values that best described most of the measurements. As mentioned in the previous sections, when we remove the cases in which the phase had extreme values, there is a linear dependance

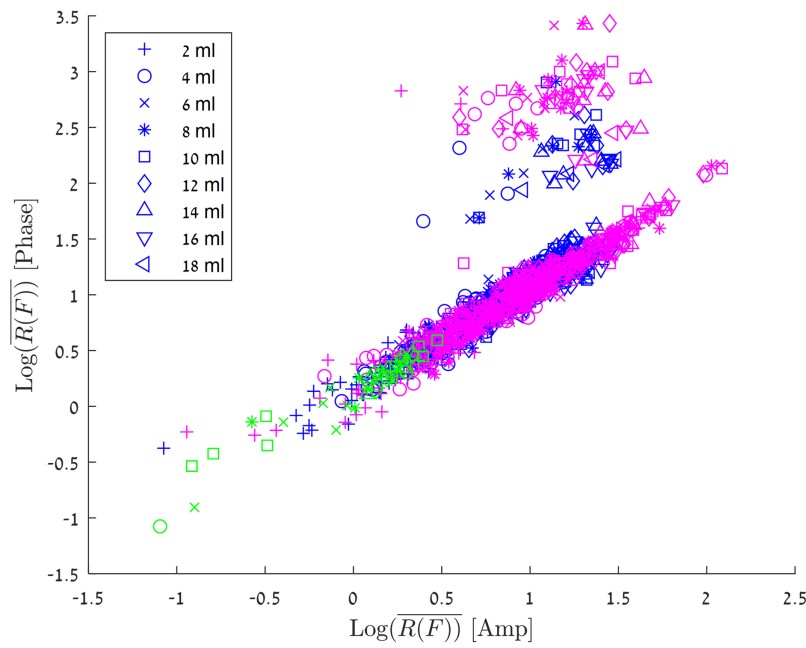

**Figure 6 Log$_{10}$ (R(F)) for the amplitude vs Log$_{10}$ (R(F)) for the phase.** The experiments in which the blood was injected in/out the balloon marked with green markers, experiments in which the blood was injected into a balloon marked with blue markers and experiments in which the blood was injected directly into the brain marked with red markers.

between the values of amplitude and phase. Therefore, we have chosen the measurements that maintain:

$$0.8 < \frac{\text{Log}_{10}\left(\overline{R(F)_{\text{phase}}}\right)}{\text{Log}_{10}\left(\overline{R(F)_{\text{amp}}}\right)} < 1.5 \tag{3}$$

At the same time, values higher than the noise values were selected:

$$\text{Log}_{10}\left(\overline{R(F)_{\text{amp,phase}}}\right) > 0.5 \tag{4}$$

To generate the model, we selected all the measurements that meet the above criteria from the measurement sets presented in Figs. 5 and 6, and we filter out each injected volume belonging to the outlier's values from the box plots of Fig. 5 (marked with a red plus). We would like to do so, to avoid creating a situation in which the estimated blood volume values will be strongly influenced by the extreme values.

We used a linear regression model with two predictors $\text{Log}_{10}\left(\overline{R(F)}\right)$ (for amplitude and phase) as follows:

$$Y = b_0 + b_1 X_1 + b_2 X_2 + b_3 X_1^2 + b_4 X_2^2 \tag{5}$$

Figure 7 visually presents the adjustment of the chosen measurement values to the linear regression model where the mean square error is 1.76 ml.

To examine the model, we ran eight additional measurements (including 9 injections from 2 to 18 ml) that were not part of the measurement set for model construction.

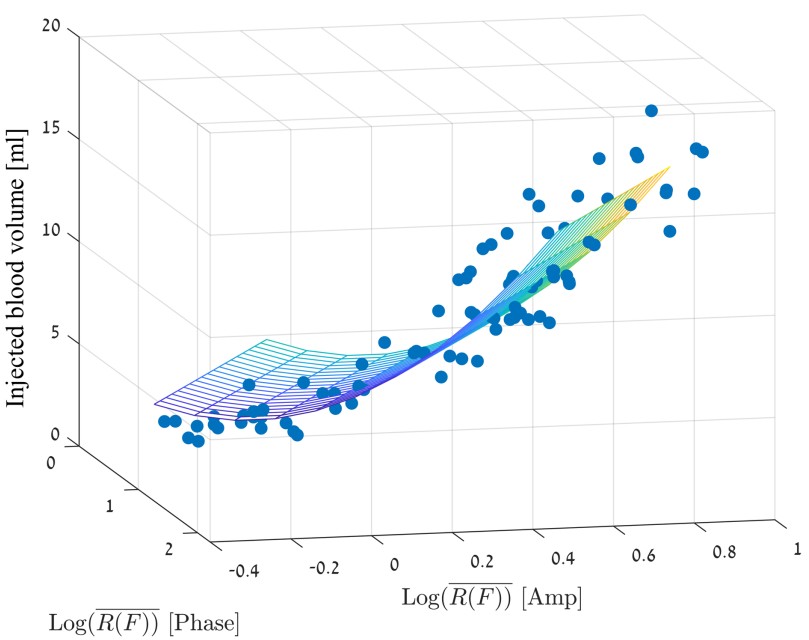

**Figure 7 Linear regression model for the injection volume based on the amp/phase values.**

The measurements for the model of the brain were different than the measurements made for the model creation, by the following parameters: We replaced the brain tissue 3 times. Since the brain structure is not uniform, the brain volume and the spatial structure introduced to the skull were slightly different for each replacement of brain tissue and as well the amount of the saline. We verified that the locations of blood injection would not be in the same places as in the model building phase. We made sure as well, to change the phantom position relative to the coil between each measurement (around half a cm for each direction) and we also took care to remove and return the coil between each measurement, so that its position will somewhat vary. Four measurements were performed when the blood was injected into a balloon, and the other four measurements were made following direct injection into the brain tissue. For each group of four measurements, two measurements were made close to the center of the brain and two closes to the phantom margin but not in the same places where the injection sites for model creation were performed.

As previously mentioned, the basis for deciding whether there has been a significant change in tissue/blood volume ratio was that $\text{Log}_{10}\left(\overline{R(F)}_{\text{amp,phase}}\right) > 0.5$. For a two ml volume difference, in 82% of the tests the algorithm values were higher than the threshold; for a four ml volume differences, about 90% of the measured values were higher than the threshold. At six ml, approximately 99% measured values were higher than the threshold, and from 8 to 18 ml, 100% of the values were higher than the threshold.

Figure 8 shows the mean and standard deviation of the estimated blood volumes after filtering out the values that do not satisfy Eqs. (3) and (4). One should note that the estimated volumes are greater than the true volume values for all cases. This phenomenon

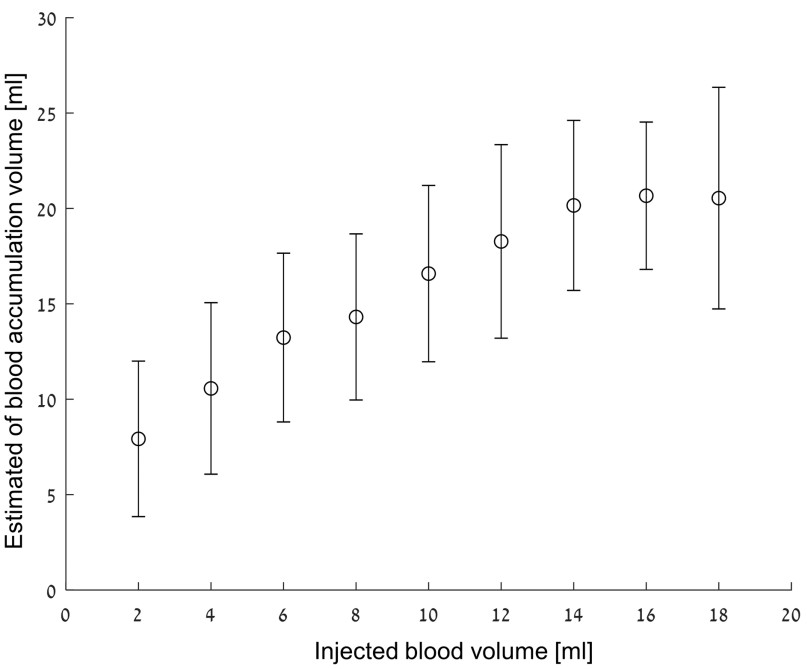

**Figure 8 Estimated volume vs injected volume.** The results are given as mean ± SD.

is understandable considering that most of the extreme values that were filtered out during model construction were high, and therefore, the volume estimation is biased upward. One can see how the error is more or less fixed (2.3 ml average for the average volumes) and a monotonous increase of the estimated volume in the 0–16 ml range. One way to interpret the figure is to evaluate what is the probability to estimate a volume Y from the analysis algorithm if volume X was actually injected. For example, when an actual injection of 12 ml blood takes place, the probability that the analysis algorithm will estimate a value between 10 and 16 ml is about 50%. However, the probability that the detection algorithm will estimate that the injection is between 10 and 16 ml if the actual injection is two ml is almost zero.

## The phantom as a human head model

Designing a human head model for the simulation of head blood flow should take into account a number of parameters, both in the area of the dielectric properties of the various head components and in the different physiological aspects.

Regarding the dielectric properties of the phantom, even though chicken red blood cells differ from mammalian blood cells both in their shape and the existence of a cell nucleus in avian blood cells (*Zentgraf, Deumling & Franke, 1969*), the dielectric properties of hen blood are similar to those of human blood in the frequency range of 100–1,000 MHz. Table 1 presents a comparison between the dielectric parameters of human blood and chicken blood, as well as dielectric data for calf and human brain.

At the beginning of each experiment (i.e., before the first blood injection), the phantom contained no blood at all—unlike a human head. However, we believe that this change

**Table 1 Dielectric parameters.** The human brain, skin and the blood dielectric properties derived from *Hasgall et al. (2018)*. Chicken blood properties derived from *Kratzenberg, Afsar & Wang (2003)*. Bovine brain properties were derived from *Gabriel, Lau & Gabriel (1996)*. PVC properties were derived from *Chung, Sabo & Pica (1982)* and saline properties derived from *Alanen (1999)*.

| Frequency | Human brain | | Human skin | | Bovine brain | | Human blood | | Chicken blood | | Saline | | Human skull | | PVC | |
|---|---|---|---|---|---|---|---|---|---|---|---|---|---|---|---|---|
| | $\varepsilon_r$ | $\sigma$ | $\varepsilon_r$ | $\sigma$ | $\varepsilon_r$ | $\sigma$ | $\varepsilon_r$ | $\sigma$ | $\varepsilon_r$ | $\sigma$ | $\varepsilon_r$ | $\sigma$ | $\varepsilon_r$ | $\sigma$ | $\varepsilon_r$ | $\sigma$ |
| 100 MHz | 89.76 | 0.79 | 72.9 | 0.49 | 94.2 | 0.80 | 76.8 | 1.23 | 80.2 | 0.92 | 120 | 1.48 | 15.28 | 0.06 | 8.5 | $<10^{-3}$ |
| 500 MHz | 53.82 | 1.08 | 49.9 | 0.72 | 47.0 | 1.10 | 63.3 | 1.38 | 65.8 | 1.41 | 61 | 1.51 | 12.95 | 0.10 | 9.1 | $<10^{-3}$ |
| 1,000 MHz | 48.85 | 1.30 | 40.9 | 0.90 | 41.6 | 1.30 | 61.1 | 1.58 | 63.9 | 1.60 | 55 | 1.56 | 12.36 | 0.15 | 9.3 | $<10^{-3}$ |

doesn't present a major impact on the experimental result since the empty phantom parts was filled with physiological saline. It can be seen in Table 1 that the dielectric values of the physiological saline are very close to that of blood. Similarly, it can be seen in Table 1 that the electrical properties of a human skull and of PVC are also very close. Since the head's skin thickness is very small (about 3 mm), and in addition, the electrical properties of human skin are very close to those of the Bovine brain—the effect of the skin layers on the signal strength is nil and can be neglected.

Another physiological aspect is the constant volume of the skull, due to the skull rigidity. This aspect is also preserved in our experiments. When we have injected the blood, the pressure in the skull has raised and in most of the cases, the brain tissue was slightly compressed towards the small openings at the base of the skull (The physiological term for this process is called Herniation). The volume of CSF (Cerebrospinal fluid) in the head varies over time depending on a large number of variables such as the blood pressure, posture change, and etc. (*Van Beek et al., 2008*) However, experiments in rabbits (*Zhao et al., 2019*) in which the change in the signal phase was monitored for 24 h, showed that the accumulation of blood in the brain has a contribution to the change in the phase 10 times greater than the contribution in the phase as a result of the normal changes in fluid in the head.

Brain activity produces electrical signals (EEG). However, the frequency range of the electrical activity in the brain is much lower (1–2,000 Hz) (*Moffett et al., 2017*) than the operating range of the system proposed in this article (100–1,000 MHz). Therefore, the electrical signals from the brain activity do not actually affect the signal transmitted/received from the measurement system.

An additional physiological aspect is blood pulsatility in the head. This aspect was examined and reported in the results section, where we tested (separately) the signal strength resulting from an injection of one ml of blood in and out the phantom. We have verified that this signal is very low and is below the algorithm threshold of detecting a significant change in blood volume in the head.

## Limitation of the study

The study assumes that the measurement setup monitors the patient while the hemorrhage is developing. In order to bring the research into clinical trial phase, two significant
problems should be resolved. These two problems are deeply related to the capabilities of the system to deal with motion artifacts.

The first topic is to find a better alternative to the Faraday cage we have used so far. In the experimental system, we used a metallic cylinder similar to a pot to avoid interference resulting from the reflection of RF waves from the environment.

This pot has a number of drawbacks. The first is the patient's discomfort as a result of attaching a massive metal block to the head. The second is the difficulty of bringing back the pot to its exact location when the coil is temporarily removed. One possible solution is the use of a mesh mask similar to those used for radiotherapy. The use of such a mask allows the coil to be positioned back to its exact location, which is convenient both for the patient and caregiver, and the configuration of this coil would reduce the patient's discomfort.

However, the mesh also has a drawback common to that of the pot. This disadvantage is the closed spatial structure, which returns the RF reflections from the boundaries back to the coil. This configuration is similar to the configuration of the two coils, in terms of the sensitivity of the system to coil displacements or the head displacements with respect to the coil. A possible solution would be to build a new coil, where the external coil is covered with a metallic material. This configuration will significantly reduce the waves transmitted to the area outside the coil and therefore significantly reduce the reflections from the environment, although it will not prevent it altogether.

The second point is the need to build a cable-free measurement system. In such a futuristic setup, the RF components should be physically connected as tightly and as close as possible to the coil, with no flexible cables between the coil and transmitter/receiver. Similarly, the sampling component should be designed to transmit the amplitude/phase values to a computer or other communication component, but will not be connected by a communication cable. These enhancements will allow maximum comfort for the patient and significantly reduce noise insertion as a result of unavoidable patient movement in the space.

Solving these two problems is expected to lead to the ability to overcome both motion artifacts arising from unavoidable movements of the patient as well as motion artifacts arising from the wearing and removal of the measurement system.

Although these two problems are very significant for the general use of this technology, it is important to note (*Davis et al., 2006*) that in many cases (over 70%) the blood volume in the brain increases by 3 h after arrival to the hospital (*Davis et al., 2006*). Therefore, if the hospital has the ability to stabilize the patient and prevent motion artifacts, clinical uses of this technology can be seen even before upgrading the present set-up.

## Cost

A new Network Analyzer can indeed cost several thousand dollars. However relative to CT or MRI which cost millions of dollars, such a solution is relatively inexpensive. In addition, the authors of this article believe that manufacturing such a device in an industrial manner so that it measures a limited and constant number of frequencies can significantly lower the price of such a device.

## CONCLUSIONS

The current study validates the algorithm accuracy based on using a phantom that has similar dielectric properties and spatial structure as those possessed by a human head. We have shown in this study that the suggested simple and inexpensive measurement system combined with the algorithm can detect slight changes in the tissue/fluid volume ratio in the head, up with the ability to detect progressive internal bleeding in the head of a minimal volume of two ml. Based on such characteristics, it may be possible to build a simple device capable of detecting and alerting, at the beginning of the increase in the volume of bleeding in the head, which may help physicians initiate early intervention to prevent further deterioration of brain injury. We also showed how the volume of blood injected into the brain can be roughly measured. However, it is clear that there is a need to improve and to test the algorithm prior to clinical trials.

## ACKNOWLEDGEMENTS

This work is based on a portion of a dissertation to be submitted by Moshe Oziel in partial fulfillment of the requirements for a PhD degree to Tel-Aviv University.

### Funding

The authors received no funding for this work.

### Competing Interests

Boris Rubinsky is an Academic Editor for PeerJ.

### Author Contributions

- Moshe Oziel conceived and designed the experiments, performed the experiments, analyzed the data, prepared figures and/or tables, authored or reviewed drafts of the paper, and approved the final draft.
- Boris Rubinsky conceived and designed the experiments, analyzed the data, prepared figures and/or tables, authored or reviewed drafts of the paper, and approved the final draft.
- Rafi Korenstein conceived and designed the experiments, analyzed the data, prepared figures and/or tables, authored or reviewed drafts of the paper, and approved the final draft.

### Data Availability

The raw measurements are available in the Supplemental Files.

### Supplemental Information

Supplemental information for this article can be found online at http://dx.doi.org/10.7717/peerj.10416#supplemental-information.

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
