# Peer review of "Detection and estimating the blood accumulation volume of brain hemorrhage in a human anatomical skull using a RF single coil"

_PeerJ, doi:10.7717/peerj.10416_

## Round 0.1 · original submission · Major Revisions

Dear Authors,

Please revise your manuscript according to the comments of the two peer reviewers. Thank You.

Reviewer 1 ·

Basic reporting

This research investigates the use of an RF coil to measure changes in the tissue/fluid volume ratio in a head phantom, simulating an intracranial hemorrhage.

The paper is well written and presents an interesting method for detection of intracranial hemorrhage. The literature review is lacking a few key studies, but is sufficient.

The article is well structured with appropriate figures and tables.

Experimental design

The research experimental design is well structured with useful and supporting methods to investigate the hypothesis. Overall the research is well designed.

Minor concerns:
An actual image of the sensing inductive coil is missing.
What is the permittivity of the phantom skull and how is it different to an actual bone?
The model does not take into account the properties of actual skull bone and skin layers.
The model does not take into account other biofluids which may change the effective permittivity such as cerebral spinal fluid shifts.

How do the investigators ensure that the changes in signal reading were not due to changes in the relative placement of the faraday cage.

The single coil will be sensitive to motion artifacts as well if the spatial symmetry surrounding the head is not maintained.

Line 246: Change "let define" to "let us define"

It would be helpful to present an S11 graph frequency sweep of the entire frequency range so that the resonant frequency response and peaks can be seen.

Validity of the findings

Volume is not being estimated, instead, the authors are only measuring change in volume. As such, the title of the paper should be reconsidered to not include volume estimation. The authors incorrectly assume the initial blood volume of the head is zero. While this may be correct in their phantom, the tissue/fluid ratio is not. Figure 3, Figure 5, Figure 7, and Figure 8 should all be revised to indicate that it is a change in blood volume and not an absolute volumetric measurement or estimation.

Additional comments

A vector network analyzer is expensive as well. It is not clear how this technology meets the primary rationale for this work to develop a low cost simple to use diagnostic tool.

Line 61: Please revise comment that the head is a closed "box" to something more anatomically correct.

Reviewer 2 ·

Basic reporting

This is an interesting study. In the study, you tried to prove the efficacy of your designed measurement system in detecting intracranial hemorrhage in a simulating model. The English of the manuscript is easy to understand. The citations of literature are proper. Photos of the experimental setup and brain model provided serve as intuitive visual evidence for readers to understand the study method.

In the conclusion, line 523, I suggest rephrasing “which may reduce the deterioration” to “which may help physicians initiate early intervention to prevent further deterioration” to make sense.

Experimental design

The use of a simple in vitro brain model with calf brain tissue, chicken blood, and anatomical skull simulating human intracranial hemorrhage is quite unique. However, in the discussion section, you need to justify the efficacy of using such an in vitro model. How much does it differ from real live brain tissue in vivo? As live brain activities generate electrical signals, this auto-generated electroencephalography interference will most likely compromise the proposed detection method.

Validity of the findings

The data and algorithm provided are robust in terms of the accuracy of such a model to detect bleeding. The goal of building such an inexpensive and safe device that can detect early stroke is of significant clinical importance.

Additional comments

I commend the authors for their unique experimental design.
If the author can justify their use of an in vitro calf brain that is devoid of electrical activities in the discussion section, the rationale of the experiment design could be further improved.

---

## Round 0.2 · accepted · Accept

Dear Authors, Thank you for your revised manuscript which has been accepted.Thanking you.

Reviewer 2 ·

Basic reporting

Changes were made based on the previous comments. It looks better now.

Experimental design

The chapter of 'The Phantom as a Human Head Model' attempted to justify the use of the phantom. I see it was moved to the Discussion section which makes sense now.

The added paragraph explaining the difference in the frequency ranges between the electrical activity in the brain and that of the operating system does sound logical.

Validity of the findings

Data and algorithm provided are robust.

Additional comments

The manuscript is improved. Good.